# Robust Privacy: Inference-Time Privacy through Certified Robustness

## Abstract

Machine learning systems can produce personalized outputs that allow an adversary to infer sensitive input attributes at inference time. We introduce Robust Privacy (RP), an inference-time privacy notion inspired by certified robustness: if a model's prediction is provably invariant within a radius-$R$ neighborhood around an input $x$ (e.g., under the $\ell_2$ norm), then $x$ enjoys $R$-Robust Privacy, i.e., observing the prediction cannot distinguish $x$ from any input within distance $R$ of $x$. We further develop Attribute Privacy Enhancement (APE) to translate input-level invariance into an attribute-level privacy effect. In a controlled recommendation task where the decision depends primarily on a sensitive attribute, we show that RP expands the set of sensitive-attribute values compatible with a positive recommendation, expanding the inference interval accordingly. Finally, we empirically demonstrate that RP also mitigates model inversion attacks (MIAs) by masking fine-grained input–output dependence. Even at small noise levels ($\sigma = 0.1$), RP reduces the attack success rate (ASR) from 73% to 4% with partial model performance degradation. RP can also partially mitigate MIAs (e.g., ASR drops to 44%) with no model performance degradation.

## 1. Introduction

Personalized machine learning systems, such as recommender engines and predictive models, can sometimes raise privacy concerns at inference time. While models that leverage rich user data can improve performance, highly specific model outputs can inadvertently serve as a side channel, enabling an observer to infer sensitive attributes of the input from the released output (Fredrikson et al., 2014; Zhao et al., 2021; Mehnaz et al., 2022; Jayaraman & Evans, 2022;

[1]Anonymous Institution, Anonymous City, Anonymous Region, Anonymous Country. Correspondence to: Anonymous Author <anon.email@domain.com>.

Preliminary work. Under review by the International Conference on Machine Learning (ICML). Do not distribute.

Mireshghallah et al., 2023). For example, if a model recommends insulin only when a user's Body Mass Index (BMI) exceeds a threshold, an adversary can infer the user's BMI range based solely on the recommendation.

This phenomenon contributes to the sense of privacy leakage at inference time. Existing privacy frameworks largely focus on protecting data during model training (e.g., differential privacy applied to the training process (Abadi et al., 2016)), leaving a gap in protections for individual inputs at inference time (Mireshghallah et al., 2023). In this paper, we address this gap by proposing Robust Privacy, an inference-time privacy notion.

Recent research in adversarial machine learning has proposed techniques (e.g., randomized smoothing (Cohen et al., 2019)) that make models certifiably robust to input perturbations. Formally, certified robustness aims to ensure that a model's prediction remains unchanged for all inputs within a neighborhood of a given input, with the neighborhood defined by a robust radius $R$ under the $\ell_p$ norm (e.g., $p \in \{1, 2, \infty\}$) (Lecuyer et al., 2019; Li et al., 2019; Cohen et al., 2019). We repurpose this notion for inference-time privacy: if an adversary cannot distinguish an input $x$ from a nearby variant $x'$ based on the model's identical prediction, then releasing predictions does not constitute a side channel that leaks fine-grained information about $x$ at inference time. We formalize this invariance-based notion of inference-time privacy as Robust Privacy.

We formally define Robust Privacy in Section 4. The robust radius $R$ quantifies the size of the indistinguishable neighborhood around an input. Larger $R$ therefore corresponds to stronger inference-time privacy. We then translate input-level invariance under Robust Privacy into an attribute-level privacy effect, which we formalize as Attribute Privacy Enhancement (APE).

In Section 5, we motivate the need for inference-time privacy in practice through a controlled recommendation task that depends primarily on a sensitive user attribute. Using BMI as the sensitive attribute, we instantiate Robust Privacy with randomized smoothing and evaluate the resulting APE effect: the range of BMI values consistent with a positive recommendation expands, reducing the precision of sensitive attribute inference from the released prediction.

In Section 6, we empirically demonstrate that Robust Privacy mitigates model inversion attacks (Kahla et al., 2022). Such attacks aim to reconstruct training data, and their effectiveness critically depends on the distinguishability of model outputs with respect to input variations at inference time. By enforcing output invariance over a neighborhood of inputs, Robust Privacy masks the fine-grained input–output dependence required for precise reconstruction, thereby limiting this inference-time attack surface. Our experimental results show that the extent of mitigation depends on the parameterization: the attack success rate (ASR) drops from 73% to 4% with partial model performance degradation, and drops from 73% to 44% with no performance degradation.

We conclude that Robust Privacy is an inference-time privacy framework for machine learning systems that (i) interact with sensitive user data, where it can mitigate sensitive attribute inference at inference time, and (ii) are trained on sensitive datasets, where it can mitigate model inversion. Robust Privacy complements existing privacy protection frameworks (e.g., Differential Privacy (Dwork et al., 2006)).

Our contributions are:

- **Robust Privacy.** We introduce Robust Privacy (RP), an inference-time privacy notion that repurposes certified robustness, providing an indistinguishability guarantee within an $R$-bounded neighborhood around each input, thereby mitigating inference-time side-channel leakage about the input through released predictions.

- **Attribute Privacy Enhancement.** We formalize Attribute Privacy Enhancement (APE) to translate input-level invariance under RP into an attribute-level privacy effect, quantifying privacy protection by the magnitude of sensitive-attribute inference-interval expansion.

- **Inference-time privacy evaluation.** We formalize the threat model (i.e., attribute inference with background knowledge) and use a controlled recommendation task, where the decision primarily depends on a sensitive attribute, to demonstrate the APE effect: RP expands the set of sensitive-attribute values consistent with a positive recommendation, reducing inference precision.

- **Mitigation of model inversion attacks.** We formalize the threat model (i.e., black-box access with hard-label outputs) and empirically show that RP mitigates model inversion attacks by masking fine-grained input–output dependence that guides iterative optimization, and evaluate the mitigation effect and model performance under different parameterizations.

## 2. Related Work

Robust Privacy is inspired by certified robustness, and our threat model is closely related to prior work on attribute inference and model inversion. We briefly review these lines of work and explain how Robust Privacy can mitigate both attribute inference and model inversion.

### 2.1. Attribute Inference Attacks

Fredrikson et al. (Fredrikson et al., 2014) introduced the concept of attribute inference: given a pharmacogenetic model for warfarin dosing, an adversary uses the model output (i.e., the predicted dosage) together with known background attributes to infer sensitive genetic markers. Their results show that model outputs can leak sensitive input attributes at inference time. Subsequent work studied attribute inference in more general ML settings, including systematic analyses of when such inference is feasible and when it reduces to standard imputation (Zhao et al., 2021; Mehnaz et al., 2022; Jayaraman & Evans, 2022). These studies typically assume that all attributes other than the target sensitive attribute are known, consistent with Scenario I in Section 3. Many attacks exploit richer (soft) model outputs (e.g., confidence vectors or regression values), while some works also consider label-only access (Mehnaz et al., 2022). In Section 5, we propose a label-only, black-box attribute inference attack and evaluate Robust Privacy as an inference-time privacy mechanism for sensitive input attributes.

While this line of work is sometimes viewed as an instance of model inversion, we treat it as attribute inference because it recovers only a subset of a query subject's attributes. In contrast, we use model inversion to refer to attacks that optimize over the input space to approximately reconstruct representative samples from the model's training distribution (e.g., a facial image).

### 2.2. Model Inversion Attacks

Model inversion attacks aim to reconstruct representative inputs that reveal information about a model's private training data. Early work typically exploits soft outputs (e.g., confidence scores) as a reconstruction signal. Building on the attribute-inference setting (Fredrikson et al., 2014), Fredrikson et al. (Fredrikson et al., 2015) formalized model inversion using confidence values and showed that recognizable face images can be recovered with black-box access. Subsequent work further studied black-box inversion in settings where the adversary has access to soft outputs. For example, Yang et al. (Yang et al., 2019) proposed to improve inversion by aligning auxiliary knowledge.

Recent attacks improve inversion quality by incorporating generative models or priors. Aïvodji et al. (Aïvodji et al., 2019) introduced GAMIN, a black-box inversion framework that jointly trains a surrogate and a generator in a generative-adversarial manner by querying the target model. Zhang et al. (Zhang et al., 2020) optimize in a GAN latent space with a public-data prior to produce reconstructions under a target

model, and Chen et al. (Chen et al., 2021) further enhance fidelity by distilling knowledge from the target model into the generator, yielding samples that better match the private distribution.

More recently, Kahla et al. (Kahla et al., 2022) showed that inversion remains possible even in a stricter label-only, black-box setting: the adversary probes small perturbations and tracks label changes to estimate a directional update signal, enabling recognizable face reconstructions of the training set.

Across these settings, a key enabler is that inference-time outputs remain sensitive to small input variations, providing an iterative optimization signal via confidence or label changes. Robust Privacy is designed to mask this inference-time signal by enforcing local output invariance in a neighborhood around the queried input.

### 2.3. Certified Robustness

Certified robustness studies whether a classifier's prediction is provably invariant within a neighborhood of a given input. Concretely, a robustness certificate for an input $x$ provides a robust radius $R$ such that the prediction remains unchanged for all inputs $x'$ with $\|x' - x\|_p \leq R$ (e.g., $p \in \{1, 2, \infty\}$). In the robustness literature, this rules out the existence of adversarial examples (Szegedy et al., 2014; Goodfellow et al., 2014; Carlini & Wagner, 2017; Madry et al., 2018) within that neighborhood. In this study, we formalize Robust Privacy via certified robustness and repurpose its output-invariance guarantee for inference-time privacy. Existing certification approaches broadly fall into two categories based on how the guarantee is obtained.

**Deterministic conservative certification.** A large class of methods provides deterministic, sound (but typically incomplete) certificates by conservatively over-approximating a network's behavior over a defined neighborhood. Representative approaches include bound-propagation methods such as CROWN (Zhang et al., 2018) and $\beta$-CROWN (Wang et al., 2021), as well as optimization-based verifiers based on convex relaxations (e.g., linear programming) or mixed-integer formulations (Ehlers, 2017; Tjeng et al., 2019; Bunel et al., 2020). These techniques certify invariance by deriving bounds on the model's outputs that hold for all perturbations in the neighborhood, which implies that the predicted class cannot change within it. In practice, the resulting robust radii are often conservative and depend on the tightness of the underlying approximation and, when applicable, the optimization budget or termination criteria.

**Probabilistic certification via smoothing.** Smoothing techniques (Cohen et al., 2019; Li et al., 2019) provide probabilistic robustness certificates by defining a smoothed

classifier that injects random noise into the input and predicts by majority vote over the base classifier's noisy evaluations. The robust radius is obtained from bounds on the resulting class probabilities, and is reported together with a user-chosen failure probability $\alpha$.

In this work, we instantiate Robust Privacy using the smoothing framework of Cohen et al. (Cohen et al., 2019), as it is a widely used baseline with standard $\ell_2$ certificates and enables direct control of the inference-time privacy level by tuning its parameters. Let $f$ denote the base classifier that maps an input $x$ to a label in $\{1, \ldots, K\}$. Randomized smoothing defines a smoothed classifier $g$ by injecting isotropic Gaussian noise $\epsilon \sim \mathcal{N}(0, \sigma^2 I)$, where $I$ is the identity matrix, into the input and predicting the most likely class: $g(x) = \arg\max_{k \in \{1, \ldots, K\}} \Pr(f(x + \epsilon) = k)$. Their robust radius is derived as:

$$R = \frac{\sigma}{2}\Big(\Phi^{-1}(\underline{p_A}) - \Phi^{-1}(\overline{p_B})\Big), \tag{1}$$

where $\Phi^{-1}$ is the inverse of the standard Gaussian CDF and $\sigma$ is the Gaussian noise scale. In practice, $\underline{p_A}$ and $\overline{p_B}$ are estimated from $N$ Monte Carlo samples at failure probability $\alpha$ as a lower/upper confidence bound on the top/runner-up class probability under the noisy predictions, respectively. This parameterization allows us to demonstrate different privacy levels in our experiments.

## 3. Threat Model

We consider a strict label-only, black-box adversary. The attacker can adaptively submit inputs to the model and observe only the returned label, without access to confidence scores, gradients, or internal model parameters. Under this general capability, we instantiate two specific attack scenarios corresponding to our evaluations.

**Scenario I: Sensitive Attribute Inference.** In this setting, the attacker targets a specific user at inference time and seeks to infer a sensitive attribute $x_1$ (e.g., BMI) from the model's predicted label.

To formalize side information, we let $x_{-1}$ denote all attributes other than the sensitive target attribute $x_1$, and assume that these remaining (non-sensitive) attributes are commonly observable, obtainable, or reasonably guessable in practical deployments. This is a standard assumption in attribute inference attacks (Fredrikson et al., 2014; Zhao et al., 2021; Mehnaz et al., 2022; Jayaraman & Evans, 2022). For example, $x_{-1}$ may include coarse profile or demographic attributes that the user discloses to access the service, as well as attributes visible from a public profile. Accordingly, we consider a side-information model in which the attacker knows $x_{-1}$ but does not know $x_1$.

We also assume the attacker can observe the user's predic-

tion result because the model output is shown to the user and may be (i) directly observed by a nearby adversary, (ii) captured through screen sharing, screenshots, or telemetry logs accessible to the adversary, or (iii) inferred from downstream actions triggered by the decision (e.g., a recommended item being displayed or a service being offered or withheld).

Given $x_{-1}$ and an observed label $y = f(x_1, x_{-1})$, the attacker's goal is to infer the value of $x_1$ or a narrow plausible range. When the model's decision depends strongly on $x_1$ and only weakly on the remaining attributes (e.g., recommending insulin primarily based on BMI), observing $y$ can yield a tight inference interval for $x_1$. This setting motivates our Attribute Privacy Enhancement (APE) analysis, where the assumption that $x_{-1}$ is known provides a conservative baseline for quantifying the attribute-level privacy enhancement induced by RP.

**Scenario II: Model Inversion Attack.** In this setting, the attacker does not target a specific sensitive attribute. Instead, the attacker aims to reconstruct representative inputs that the model predicts as a chosen target class (i.e., a model inversion attack), thereby revealing visual or semantic characteristics of the private training data.

Under the label-only constraint, the attacker iteratively queries the model and updates a synthetic input based solely on observed labels. For example, in the label-only inversion attack of (Kahla et al., 2022) that we evaluate, the attacker estimates a directional update signal by probing the model with small perturbations to the input and observing whether the predicted label changes. The attacker's objective is to navigate the input space toward regions where the model consistently predicts the target label. By reaching such regions, the attacker recovers inputs that expose class-specific features present in the training data.

## 4. Robust Privacy

In this section, we formalize the notion of Robust Privacy and show how it induces privacy enhancement at the attribute level.

**Robust Privacy (RP).** We first formalize the notion of Robust Privacy, which characterizes input-level output invariance as a privacy guarantee.

**Definition 1** (Robust Privacy). *Let $f : \mathcal{X} \to \mathcal{Y}$ be a predictive model, and let $\| \cdot \|_p$ denote an $\ell_p$ norm on $\mathcal{X}$. For an input $x \in \mathcal{X}$, we say that $x$ enjoys R-Robust Privacy under $f$ if for every $x' \in \mathcal{X}$ satisfying*

$$\|x' - x\|_p \leq R,$$

*the model output remains invariant, i.e.,*

$$f(x') = f(x).$$

*The non-negative scalar $R$ is called the robust radius at $x$.*

Unless otherwise specified, we consider standard $\ell_p$ norms with $p \in \{1, 2, \infty\}$, which are commonly used in robustness analysis (Cohen et al., 2019; Li et al., 2019). Definition 1 characterizes Robust Privacy as a deterministic property. Depending on how RP is instantiated, the resulting guarantee can be either deterministic or probabilistic. When the guarantee is provided by deterministic robustness verification techniques such as bound propagation (e.g., CROWN (Zhang et al., 2018) and $\beta$-CROWN (Wang et al., 2021)), output invariance holds deterministically. In contrast, Robust Privacy can also be instantiated using smoothing techniques (Cohen et al., 2019; Li et al., 2019), in which case the guarantee is probabilistic: the robust radius $R$ is associated with a failure probability $\alpha$, meaning that $R$ is a valid certificate for the output invariance guarantee with probability at least $1 - \alpha$.

**Interpretation.** Robust Privacy reinterprets certified robustness through a privacy lens. For a given input $x$, RP implies that the entire $\ell_p$-ball

$$\mathcal{B}_p(x, R) \triangleq \{x' \in \mathcal{X} : \|x' - x\|_p \leq R\}$$

is indistinguishable under the model output, i.e., $\mathcal{B}_p(x, R) \subseteq f^{-1}(f(x))$. Thus, observing $f(x)$ cannot distinguish fine-grained local variations within $\mathcal{B}_p(x, R)$ around the original input $x$.

Formally, Robust Privacy ensures that

$$f(x') = f(x) \quad \text{for all } x' \in \mathcal{B}_p(x, R).$$

Thus, given an observed output $y = f(x)$, the output is compatible with every $x' \in \mathcal{B}_p(x, R)$; hence, based solely on the output, an adversary cannot distinguish $x$ from any such neighbor, and the released predictions cannot serve as a side channel for input privacy leakage at inference time.

**Attribute Privacy Enhancement (APE).** Building on Robust Privacy, we formalize an attribute-level privacy notion that reduces the precision of an adversary's inference about a sensitive attribute.

**Definition 2** (Attribute Privacy Enhancement (APE)). *Consider a model $f : \mathcal{X} \to \mathcal{Y}$ and an input $x = (x_1, x_{-1})$, where $x_1 \in \mathbb{R}$ denotes a sensitive target attribute and $x_{-1}$ represents all remaining attributes. For a fixed context $x_{-1}$, define the induced function $f_{x_{-1}}(x_1) \triangleq f(x_1, x_{-1})$. Given an observable output $y$, define the baseline inference set as the set of all sensitive attribute values compatible with $y$:*

$$I_y \triangleq \{z \in \mathbb{R} : f_{x_{-1}}(z) = y\}.$$

*Let $R_z$ denote the robust radius at $(z, x_{-1})$, with $R_z = 0$ if certification abstains. The APE-expanded inference set is defined as*

$$I_y^{(R)} \triangleq \bigcup_{z \in I_y} [z - R_z, z + R_z].$$

*If $I_y^{(R)}$ is an interval, we refer to it as the APE-expanded inference interval.*

**Interpretation.** We provide several remarks to clarify the mechanism and implications of Definition 2.

*Inference set $I_y$.* For a fixed context $x_{-1}$, the set $I_y$ characterizes the exact set of sensitive values $x_1$ that map to the observed output $y$.

*Mechanism of Expansion.* Suppose the true sensitive value is $z \in I_y$. Robust Privacy at $(z, x_{-1})$ guarantees that all inputs within the robust radius $R_z$ produce the same output $y$; consequently, holding $x_{-1}$ fixed, $z$ is indistinguishable from any value in $[z - R_z, z + R_z]$ under the released prediction. Aggregating this certified local indistinguishability across all candidates $z \in I_y$ yields the expanded set $I_y^{(R)}$.

*Simplified form under a uniform lower bound.* If $I_y$ is an interval and the robust radius along it is uniformly lower bounded, i.e., $R_z \geq R$ for all $z \in I_y$, then $I_y^{(R)} \supseteq I_y \oplus [-R, R]$. This is a simplified but conservative lower bound on the expansion. In particular, $(a, b)$ expands to $(a - R, b + R)$, and $[a, +\infty)$ expands to $[a - R, +\infty)$.

*Assumption on side information.* Our formulation assumes $x_{-1}$ is fixed and known to the adversary. This is a standard assumption in attribute inference attacks (Fredrikson et al., 2014; Zhao et al., 2021; Mehnaz et al., 2022; Jayaraman & Evans, 2022), and provides a conservative baseline for quantifying the privacy effect induced by Robust Privacy. Geometrically, fixing $x_{-1}$ restricts the feasible input space to a 1D subspace; under standard norms (e.g., $\ell_2$), the robust radius effectively bounds the variation along this specific dimension. Thus, $I_y^{(R)}$ provides a conservative estimate of the privacy gain attributable to Robust Privacy.

## 5. Experiments: Inference-Time Privacy

In this section, we demonstrate inference-time privacy enabled by Robust Privacy (RP) through a controlled recommendation task that depends primarily on a sensitive user attribute. We consider a medicine recommendation task in which a model recommends insulin primarily based on a user's BMI, together with additional health-related features. This setting allows us to evaluate how the APE effect induced by RP reduces the precision with which a sensitive attribute can be inferred from the model's outputs, by expanding the attribute's inference interval.

**Experimental Setup.** We use the Medical Insurance Cost Prediction dataset (Krishnathalla, 2025) ($100\,000$ records), where categorical features are one-hot encoded and numeric features are standardized. The recommendation label is defined as $1$ if the user's BMI exceeds the 90th percentile of the training set ($B = 33.4$), and $0$ otherwise. We train a two-layer neural network (hidden size 64, ReLU) on these binary labels with a fixed $60/40$ train/test split. To enforce reliance on the sensitive attribute (and thereby create a realistic risk of attribute leakage from released predictions), we add an $\ell_1$ penalty to the first-layer weights corresponding to all non-BMI input dimensions, encouraging the model to assign larger relative weight to BMI. All experiments were conducted on an NVIDIA A100-SXM4-40GB GPU.

We apply the randomized smoothing implementation of (Cohen et al., 2019) to instantiate Robust Privacy. Randomized smoothing defines a smoothed classifier by injecting random noise into the input and predicting by majority vote over the base classifier's noisy evaluations. For the smoothed classifier, we use $N = 1000$ Monte Carlo samples for both prediction and certification, with noise scales $\sigma \in \{1, 2, 3\}$ and failure probability $\alpha = 0.01$. As randomized smoothing provides probabilistic guarantees via confidence bounds, the smoothed classifier may abstain from returning a prediction when it does not have enough confidence, and the implications of this choice are discussed in Section 7.2. For each input, the smoothed classifier reports a robust radius $R$ with failure probability $\alpha$, and this radius defines a prediction-invariant neighborhood centered at the original input $x$.

We demonstrate inference-time privacy using the Attribute Privacy Enhancement (APE) analysis (Definition 2). To empirically verify sensitive-attribute inference-interval expansion, we craft an augmented evaluation set derived from positively predicted test samples ($D_{t,1}$). For each record in $D_{t,1}$ with original BMI $b$, we keep the original record unchanged and generate additional perturbed inputs by varying only the BMI attribute (keeping all other attributes fixed). Specifically, we overwrite the BMI value with a left-of-threshold trajectory anchored at the threshold $B$:

$$b' = B - js, \quad j \in \{0, 1, \ldots, J\},$$

where $s = 0.01$ and $J = 500$ such that $B - Js = 28.4$ (i.e., spanning $[28.4, 33.4]$). This augmentation concentrates evaluation on the privacy-critical region left of the threshold, while still retaining each record's original BMI $b$. By monitoring model predictions on these perturbed inputs, we evaluate whether positive recommendations extend to BMI values below $B$, thereby demonstrating an RP-induced expansion of the sensitive-attribute inference interval.

**Results.** We collect the predictions of both the base and smoothed classifiers on the augmented set described above. Whenever a model outputs a positive label, we record the

corresponding BMI value. Figure 1 presents the resulting BMI distributions, and Table 1 reports the corresponding robust radii and abstention rates under varying $\sigma$.

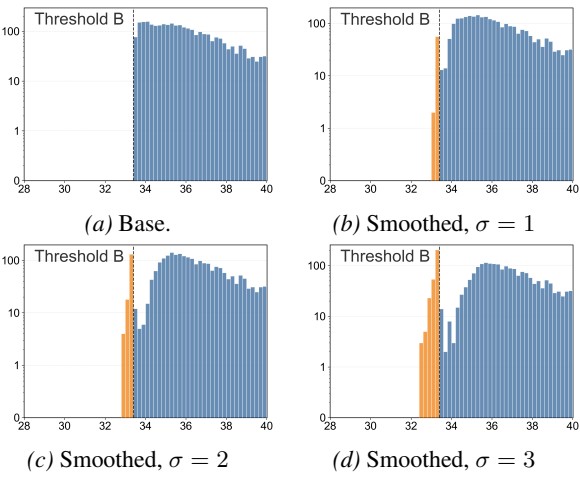

*(a) Base.*      *(b) Smoothed, $\sigma = 1$*

*(c) Smoothed, $\sigma = 2$*      *(d) Smoothed, $\sigma = 3$*

*Figure 1.* BMI distribution (x-axis) of inputs predicted as 1 (y-axis: count on a log scale) by the base classifier and by smoothed classifiers with $\sigma \in \{1, 2, 3\}$ and $N = 1000$. The vertical dashed line denotes the threshold $B$; blue/orange bins to the right/left of $B$ correspond to BMI values above/below the threshold. For the base classifier, positive predictions concentrate to the right of $B$, so observing label 1 enables a sharp inference that BMI $> B$. With Robust Privacy, positive predictions extend below $B$, consistent with the RP-induced APE effect: a positive prediction is compatible with BMI values below $B$, expanding the inference interval. Larger $\sigma$ yields a more pronounced leftward extension, indicating increased uncertainty in sensitive-attribute inference. The histogram bin width is 0.2.

As shown in Figure 1, the base classifier's positively predicted inputs are concentrated on the right side of $B$, so observing label 1 (i.e., insulin recommendation) enables a sharp inference that BMI $> B$. In contrast, the smoothed classifier classifies a fraction of inputs on the left side of $B$ as positive. This illustrates the RP-induced APE effect: upon observing a positive label, an adversary obtains a less precise inference of BMI (e.g., BMI $> B - 1$) when the model is protected with Robust Privacy, i.e., the sensitive attribute's inference interval is expanded.

As $\sigma$ increases from 1 to 3, the leftward expansion becomes more pronounced, with the observed expansion increasing from 0.4 to 1.0. This trend aligns with the increase in average robust radii from 0.56 to 0.65 reported in Table 1 (cf. Equation 1). The gap between the observed expansion and the average robust radius is expected: certifications of randomized smoothing are known to be conservative, and thus tend to underestimate the inference-interval expansion observed empirically. Moreover, we emphasize that the magnitude of the leftward expansion (i.e., how much the inference interval expands), rather than the absolute count below $B$, is the primary indicator of APE, because the ab-

solute count also depends on the sampling density used to construct the augmented evaluation set.

*Table 1.* Accuracy, abstention rate, and average robust radius $R$ (Avg. $R$) of the base classifier and smoothed classifiers for $\sigma \in \{1, 2, 3\}$ with $N = 1000$ on the Medical Insurance Cost Prediction test set. For smoothed classifiers, Avg. $R$ is computed over test samples for which the smoothed classifier makes a positive prediction. The base classifier neither abstains nor provides certificates, so its abstention rate and Avg. $R$ are reported as –.

| Model | Accuracy | Abstention Rate | Avg. $R$ |
|---|---|---|---|
| Base | 100% | – | – |
| Smoothed, $\sigma = 1$ | 96.36% | 3.64% | 0.56 |
| Smoothed, $\sigma = 2$ | 92.76% | 7.23% | 0.63 |
| Smoothed, $\sigma = 3$ | 89.04% | 10.96% | 0.65 |

We also observe that, on the right side of $B$ near the decision boundary, the smoothed classifiers assign fewer inputs to the positive class than the base classifier. This reduction is expected: when an input lies near the decision boundary, the smoothed classifiers may not have sufficient confidence to certify a prediction, resulting in an ABSTAIN outcome (Table 1), which reduces the density of observable positive predictions. We further study how $\alpha$ and $N$ influence the Attribute Privacy Enhancement effect in Appendix A.

## 6. Experiments: Mitigation against Model Inversion Attacks

In this section, we empirically demonstrate the efficacy of Robust Privacy in mitigating model inversion attacks (MIAs). These attacks aim to reconstruct training data by iteratively optimizing a synthetic input toward regions of high target-class confidence or prediction stability, using fine-grained input–output dependence at inference time as an optimization signal. Specifically, we evaluate defense against label-only model inversion attacks (Kahla et al., 2022), which rely on prediction changes under local input perturbations to estimate the optimization direction and guide the iterative reconstruction.

Robust Privacy enforces output invariance within a robust radius $R$ around an input. This inference-time privacy masks the fine-grained input–output dependence that model inversion attacks rely on to estimate an optimization direction. As illustrated in Figure 2, by making predictions constant within the $R$-bounded neighborhood, Robust Privacy deprives the attacker of this directional signal at inference time, preventing convergence to a stable target-class region.

**Experimental Setup.** We conduct the experiments on an NVIDIA A100-SXM4-40GB GPU. We apply the randomized smoothing implementation of (Cohen et al., 2019) to

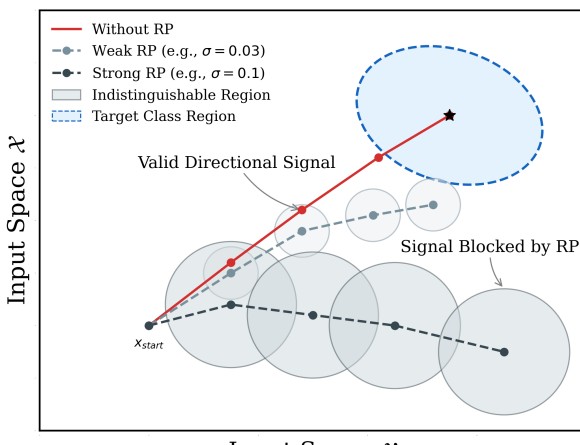

*Figure 2.* Mechanism of Robust Privacy (RP) against model inversion attacks. Without RP (red solid line), the attacker exploits local prediction changes to estimate the optimization direction and steer updates toward the target region. With RP (dashed lines), predictions remain invariant within a robust radius $R$ (influenced by the noise scale $\sigma$ and sampling size $N$). This invariance masks the input–output dependence used to estimate the local optimization direction, disrupting the attacker's iterative reconstruction process. Consequently, a small $\sigma$ (weak RP, light-grey dashed line) may still allow partial convergence to the target region, whereas a large $\sigma$ (strong RP, dark-grey dashed line) results in significant path deviation and attack failure.

instantiate Robust Privacy as a defense against the label-only model inversion attack proposed by (Kahla et al., 2022).

Randomized smoothing defines a smoothed classifier that injects random noise into the input and predicts by majority vote over the base classifier's noisy evaluations; it typically certifies the predicted label using confidence bounds. As a result, the smoothed classifier is probabilistic and may abstain from returning a prediction when it does not have enough confidence. In this experiment, we focus on the empirical mitigation effect of RP against MIAs; thus, we adopt an *always-return-a-label* protocol: the model returns a class label for every query (i.e., via majority vote under smoothing, without abstention). The implications of this choice are discussed in Section 7.2.

For the attack, we follow the recommended parameter settings from (Kahla et al., 2022), using FaceNet64 as the target model and a separate FaceNet (Schroff et al., 2015) classifier as the evaluator. Both models are trained to classify the same 1000 private identities from CelebA (Liu et al., 2015), yet they employ different architectures to ensure the evaluation reflects semantic recovery rather than overfitting.

We perform targeted model inversion on target classes with indices 0–99. The attack is conducted in two modes: (i) without Robust Privacy as a baseline to evaluate attack effectiveness, and (ii) with Robust Privacy under different noise scales $\sigma \in [0.01, 0.1]$ and sampling sizes $N \in [10, 100]$. We also evaluate the accuracy of the base and smoothed classifiers on a fixed set of 1000 CelebA images (one per private identity), where for each identity we select the image that achieves the highest FaceNet64 confidence score for that identity.

**Results.** Figure 3 summarizes both the attack success rate (ASR) and the prediction accuracy under different noise scales $\sigma \in [0.01, 0.1]$ and sampling sizes $N \in \{10, 100\}$. The baseline model (without Robust Privacy) achieves an ASR of 73%. RP reduces ASR as $\sigma$ increases: at $\sigma = 0.1$ and $N = 100$, ASR drops to 4% while the model still maintains non-trivial accuracy. Notably, for a fixed $\sigma$, increasing $N$ from 10 to 100 slightly improves the smoothed classifier's accuracy while further lowering the ASR against it, suggesting that stronger RP instantiations more reliably disrupt the attacker's directional signal, even as the smoothed classifier's prediction accuracy improves.

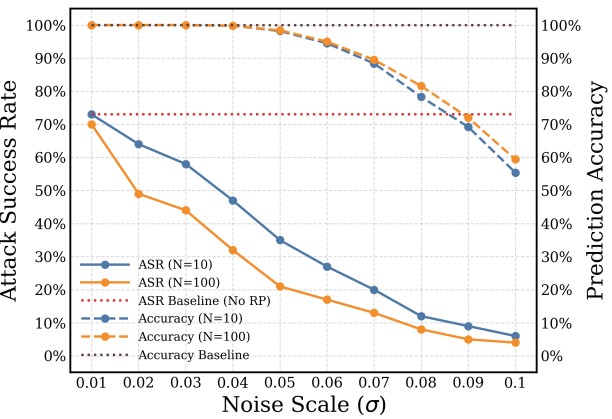

*Figure 3.* Model inversion under a label-only, black-box interface (Kahla et al., 2022): attack success rate (ASR; solid lines, left y-axis) and prediction accuracy (dashed lines, right y-axis) for the base classifier and smoothed classifiers across noise scales $\sigma \in [0.01, 0.1]$ with sampling sizes $N \in \{10, 100\}$. Horizontal dotted lines denote the base model baselines (ASR = 73%, accuracy = 100%). Enabling RP reduces ASR monotonically as $\sigma$ increases; at $\sigma = 0.1$ and $N = 100$, ASR drops to 4% while accuracy remains 59%. At $\sigma = 0.03$ and $N = 100$, accuracy remains 100% while ASR drops to 44%, demonstrating partial mitigation without compromising model performance. Increasing $N$ improves the smoothed classifier's accuracy while further lowering ASR against the smoothed classifier, highlighting that Robust Privacy mitigates MIAs by enforcing inference-time privacy rather than by degrading the model's performance.

Moreover, at $\sigma = 0.03$ and $N = 100$, the smoothed classifier retains 100% accuracy while ASR drops from 73% to 44%. This result demonstrates partial mitigation without compromising model performance, highlighting that Robust Privacy mitigates MIAs by enforcing inference-time privacy rather than by degrading the model's performance.

## 7. Discussion

In this section, we discuss practical considerations that arise when implementing and interpreting Robust Privacy (RP).

### 7.1. Tuning RP: Privacy, Utility, and Cost

Privacy, utility, and cost are closely intertwined when instantiating RP. The noise scale $\sigma$ and sampling size $N$ shape the privacy–utility balance: increasing $\sigma$ enhances privacy but can reduce accuracy. In Section 6 with $N = 100$, accuracy remains $100\%$ while ASR drops from $73\%$ to $44\%$ at $\sigma = 0.03$; at $\sigma = 0.1$, though ASR drops further to $4\%$, accuracy also drops to $59.4\%$ (Figure 3). Notably, privacy and utility do not always conflict: increasing $N$ improves accuracy while further lowering ASR, as larger $N$ typically results in a larger input neighborhood for output invariance, leading to stronger RP-induced inference-time privacy.

Larger $N$ increases inference cost, but usually improves the privacy enhancement effect. For example, we use $N = 1000$ in Section 5 to demonstrate an observable APE effect under a stringent certification configuration. However, a smaller size ($N = 10$) in Section 6 can still demonstrate low-cost but strong empirical mitigation (e.g., ASR drops to $6\%$ at $\sigma = 0.1$).

### 7.2. Handling Abstention in Our Evaluations

Smoothed classifiers may return an ABSTAIN outcome, denoted by $\perp$, when they cannot decide on a prediction with sufficient confidence. Because $\perp$ is an observable outcome, it must be handled explicitly in our evaluations. We adopt different strategies for $\perp$ in our two experiments, aligned with the distinct effects they are intended to demonstrate.

**Inference-time privacy: $\perp$ is essential for rigor.** Our first experiment aims to demonstrate the APE effect, i.e., the expansion of the sensitive-attribute inference interval. This effect is probabilistic when RP is instantiated with randomized smoothing: the interval expansion is reported with a prescribed failure probability $\alpha$. Here, $\perp$ corresponds to cases where confidence is insufficient to certify a prediction. Therefore, omitting $\perp$ and returning a prediction anyway would overstate the privacy effect. Accordingly, we retain $\perp$ in the APE evaluation and report $\alpha$ alongside the observed sensitive-attribute interval expansion, making explicit that the reported effect is certified with probability at least $1 - \alpha$.

**Model inversion mitigation: empirical evaluation.** Our second experiment evaluates the extent to which Robust Privacy (RP) mitigates model inversion attacks by disrupting the fine-grained input–output dependence that the attacker exploits to guide iterative reconstruction. In this setting, allowing $\perp$ can yield a trivial defense: if the model frequently returns $\perp$, the attack may fail simply because the attacker receives insufficient feedback, rather than because RP induces output invariance in neighborhoods of the input. Since our goal is to empirically characterize mitigation attributable to RP-induced invariance itself (rather than to abstention), we adopt an *always-return-a-label* protocol for the inversion experiments: the model returns a class label for every query (e.g., via majority vote under smoothing, without abstention). This choice ensures that the attack success rate reflects the difficulty of navigating the input space under Robust Privacy, rather than the presence of query rejections.

### 7.3. Robust Privacy vs. Differential Privacy

While Robust Privacy (RP) and Differential Privacy (DP) (Dwork et al., 2006) both support privacy-preserving machine learning, they operate at different stages and enforce distinct notions of indistinguishability. DP ensures statistical indistinguishability over neighboring training datasets, limiting the influence of any single record on the model and primarily preventing leakage of training data information (Abadi et al., 2016). In contrast, RP provides geometric output invariance in the input space at inference time, masking fine-grained sensitive attribute variations of the input from the released prediction. Thus, DP and RP offer protection at training and inference, respectively.

### 7.4. Why RP Mitigates MIAs

RP enforces inference-time privacy to mitigate MIAs. RP guarantees output invariance within a robust radius $R$ around an input at inference time. This inference-time privacy masks the fine-grained input–output dependence that MIAs rely on to estimate an optimization direction, thereby mitigating MIAs. DP-SGD (Abadi et al., 2016) can also mitigate MIAs by reducing memorization of training data. However, (Zhang et al., 2020) shows that MIAs can still reconstruct class-representative features from DP-trained models.

## 8. Conclusion

We present Robust Privacy (RP), an inference-time privacy notion inspired by certified robustness. RP reinterprets a robust radius $R$ at $x$ as an inference-time indistinguishability guarantee, so observing the released prediction cannot distinguish $x$ from inputs within distance $R$ of $x$ (e.g., under the $\ell_2$ norm). Building on RP, we introduce Attribute Privacy Enhancement (APE) to characterize an attribute-level privacy effect, and demonstrate sensitive-attribute inference interval expansion in a controlled recommendation task. Finally, we empirically show that RP mitigates model inversion attacks by masking the input–output dependence that provides an optimization signal for iterative reconstruction. Overall, our results demonstrate that robustness mechanisms can be repurposed for inference-time privacy.

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

## A. Ablation Study on the APE Evaluation

Figure 4 extends the APE experiment in Figure 1 by varying two smoothing-related parameters while fixing the noise scale to $\sigma = 3$: the Monte Carlo sampling size $N$ used to form the smoothed prediction, and the failure probability $\alpha$ that controls the strictness of the certification procedure (and thus the likelihood of abstention). This ablation helps contextualize the observed APE effect in terms of two practical aspects of randomized smoothing: statistical uncertainty due to finite sampling and confidence calibration via $\alpha$.

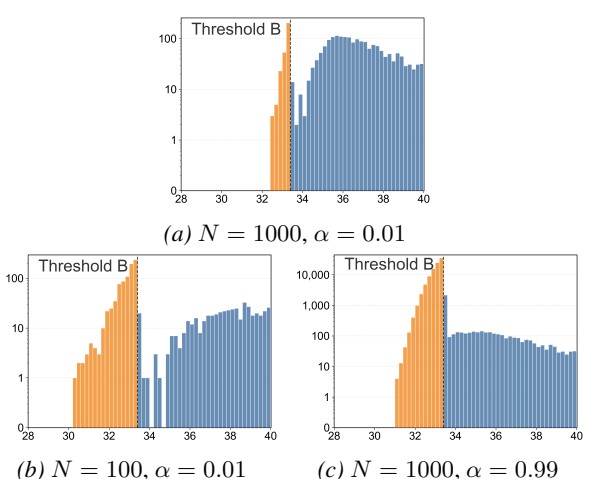

*(a) $N = 1000$, $\alpha = 0.01$*

*(b) $N = 100$, $\alpha = 0.01$*     *(c) $N = 1000$, $\alpha = 0.99$*

*Figure 4.* BMI distribution (x-axis) of inputs predicted as 1 (y-axis: count on a log scale) by smoothed classifiers with $\sigma = 3$, $N \in \{100, 1000\}$, and $\alpha \in \{0.01, 0.99\}$. The vertical dashed line denotes the threshold $B$; blue/orange bins to the right/left of $B$ correspond to BMI values above/below the threshold. With Robust Privacy, positive predictions extend below $B$, consistent with the APE interpretation: a positive prediction is compatible with BMI values below $B$ due to an expanded sensitive-attribute inference interval. This leftward extension is more pronounced with smaller $N$ and larger $\alpha$. The histogram bin width is 0.2.

Across settings, the APE effect remains: we observe positively predicted inputs extending below the threshold $B$, demonstrating that a positive recommendation can be compatible with BMI values below $B$. The magnitude of this leftward expansion varies with $(N, \alpha)$. With a smaller sampling size (e.g., $N = 100$), Monte Carlo uncertainty increases, which can make the majority-vote prediction less stable and yields more observable positive outcomes below $B$. Similarly, a larger $\alpha$ relaxes the certification confidence requirement and reduces abstentions, allowing more marginal cases to appear as positive predictions below $B$. Both effects can make the interval expansion more pronounced.

Importantly, these changes reflect a trade-off rather than a strictly stronger privacy guarantee. A larger $\alpha$ corresponds to a weaker statistical confidence requirement, and a smaller $N$ increases estimation uncertainty. Thus, the additional

mass observed below $B$ under these settings should be interpreted together with the corresponding uncertainty and abstention behavior. For the main results, we adopt conservative parameters (e.g., $N = 1000$, $\alpha = 0.01$) to present APE effects under a stringent certification configuration.

