# OpenReview forum: "Robust Privacy: Inference-Time Privacy through Certified Robustness"
_ICML.cc/2026/Conference — Submitted to ICML 2026_

### Official Review · Reviewer_ZDkB · 2026-03-01

**Soundness:** 2
**Presentation:** 2
**Significance:** 3
**Originality:** 3
**Overall Recommendation:** 3
**Confidence:** 3

**Summary:**

This paper introduces Robust Privacy (RP), a new framework to protect sensitive input attributes during model inference. Inspired by certified robustness, the authors argue that if a model’s prediction does not change within a certain radius ($R$) around an input, then an attacker who only sees the output cannot accurately recover the exact input or its detailed attributes.
The method is tested in two main settings: Attribute Privacy Enhancement (APE) on a medical insurance tabular dataset (protecting BMI values). Model Inversion Attack (MIA) defense under a strict label-only black-box setting.

**Compliance With Llm Reviewing Policy:**

Affirmed.

**Key Questions For Authors:**

Questions:
Regarding Definition 2, it is unclear what is meant by “define the baseline inference set as the set of all sensitive attribute values compatible with y.”
Does this mean all sensitive attribute values that produce the same label y? Or does it mean different possible values of one sensitive attribute? I think I understand the authors’ intention, but the definition is vague and needs clearer explanation.

**Limitations:**

Above

**Strengths And Weaknesses:**

Strength:
1. The paper builds a clear and elegant connection between two different research areas: adversarial robustness and data privacy. This idea is novel and could inspire more work in the privacy community. (However, I have some concerns about the definitions, see below.)

2. The experiments cover two different application settings, which helps show the potential flexibility of the framework.


Weakness:

1. The paper emphasizes “Inference-Time Privacy,” a term that is often linked to serious vulnerabilities in Large Language Models and generative AI (for example, extracting sensitive prompt information through API queries). However, all experiments are limited to low-dimensional tabular data and standard classification tasks. The authors would be encouraged to test the method on at least one simple NLP task or an LLM inference scenario (for example, sensitive attribute inference in text).

2. In the APE experiments, the authors show that the possible interval of sensitive attribute values (BMI) becomes larger when RP is applied. However, expanding the interval is only a mechanism, not direct proof that the defense works. To better demonstrate effectiveness, the authors should train an attack model and show that the attacker’s reconstruction error increases when RP is used.

3. The paper only compares RP with an unprotected baseline model. To show that RP is competitive, it should also be compared with other inference-stage privacy methods, such as applying Differential Privacy.

---

> ### Author Rebuttal · Authors · 2026-03-28
>
> $\color{#2F80ED}{\text{Response to W1}}$
>
> We will add a future-direction discussion, with a preliminary experiment, in the Discussion section to clarify how RP can extend to LLM settings:
>
> A natural extension of Robust Privacy is Confined Robust Privacy (CRP), which can be viewed as a sensitive-subspace version of RP: it perturbs only the sensitive part of the input rather than the full input. In an LLM setting, this means adding noise only to the embeddings of sensitive tokens, while leaving the remaining context unchanged. The goal is to protect sensitive content while preserving the surrounding semantics and utility.
>
> As a preliminary demonstration, we describe an interpretation experiment based on text reconstruction: we use Meta's encoder-decoder model bart-base to reconstruct the original input text, with Gaussian noise added only to the embeddings of sensitive tokens. Initial observations show that this can obscure the exact sensitive content while preserving the main sentence meaning; for example, a prompt such as “My allergy is penicillin” can be entirely reconstructed when no noise is added, but be reconstructed only as “My allergy is” when the embedding of “penicillin” is perturbed with noise.
>
> $\color{#2F80ED}{\text{Response to W2}}$
>
> We will add the following first paragraph in the APE experiment section and the second paragraph in the Discussion section to address the concern:
>
> The APE experiment is designed to evaluate the Attribute Privacy Enhancement effect, namely the expansion of the sensitive-attribute inference interval under the released prediction. The experimental results in Figure 1 demonstrate this exact effect.
>
> For the Attribute Inference Attack, training an attack model and reporting its attribute reconstruction error would also be a valuable additional demonstration of RP’s protection for sensitive attributes, and we leave this for future work. For now, the model inversion experiments (Figure 3) provide complementary evidence that RP can increase reconstruction error of data (e.g., facial images) in model inversion attacks.
>
> $\color{#2F80ED}{\text{Response to W3}}$
>
> To address the concern (similar to that of KQ2, Reviewer mY9D), we will add the following to the Discussion section:
>
> We use the unprotected model as the baseline because it provides the reference point for isolating the mitigation effect introduced by RP relative to the original classifier. Applying Differential Privacy at inference stage typically means adding noise to the output, which randomly perturbs the released prediction itself and therefore obviously compromises model utility. In contrast, RP instantiated with randomized smoothing uses multiple noisy versions of the input for one prediction and aggregates those predictions, so the final released prediction is stabilized rather than randomly perturbed.
>
> $\color{#2F80ED}{\text{Response to KQ1}}$
>
> We will rephrase Definition 2 to make this explicit:
>
> Given an observed output $y$, define the baseline inference set $I_y$ as the set of all possible values of the sensitive attribute that, with all other attribute values $x_{-1}$ fixed, produce the same output $y$

---

> > ### Author Rebuttal · Reviewer_ZDkB · 2026-04-01
> >
> > Thank you for the detailed rebuttal. I have read the responses carefully, and while I appreciate the interesting direction and the additional clarifications you propose, my overall assessment remains unchanged, as the paper still feels not yet ready for full acceptance.

---

### Official Review · Reviewer_wJUt · 2026-03-09

**Soundness:** 2
**Presentation:** 2
**Significance:** 2
**Originality:** 2
**Overall Recommendation:** 2
**Confidence:** 2

**Summary:**

The paper introduces Robust Privacy (RP), a framework that repurposes Certified Robustness (specifically randomized smoothing) to provide privacy guarantees at inference time. The core argument is that if a model's prediction is provably invariant within an $l_2$ radius $R$ around an input $x$, then any input within that neighborhood is indistinguishable based on the output. The authors further propose Attribute Privacy Enhancement (APE) to map this input-level geometric invariance to the protection of specific sensitive attributes.

**Compliance With Llm Reviewing Policy:**

Affirmed.

**Key Questions For Authors:**

1.	“Inference-time” is a really odd combination. Do you mean inference-stage?
2.	Privacy typically concerns whether data is present in the training set or the leakage of sensitive attributes, whereas robustness focuses on resistance to perturbations. If a model is insensitive to inputs within a certain radius, it might simply be because the model has "learned poorly" or is "oversmoothed" in that region; this does not necessarily imply that it protects user privacy. So can the authors explain more about the relationship between local robustness and privacy?
3.	In high-dimensional data like images or medical records, what does an $l_2$ radius of $R$ actually represent in terms of privacy?
4.	Why is this approach superior to simply adding noise to the output or using a DP-trained model?

**Limitations:**

The authors should argue more about the “equivalence” between local invariance and privacy.

**Strengths And Weaknesses:**

•	Soundness: The mathematical connection to randomized smoothing is technically correct, and the use of certification to provide a "hard" guarantee is theoretically interesting. But the authors equates "local robustness to input perturbations" with "attribute-level privacy", which is not very convincing.
•	Presentation: This paper is structured logically.
•	Significance: This paper offers a post-hoc way to "claim" privacy for models that are already robustly trained without needing DP-SGD during training.
•	Originality: The mapping of a robustness certificate to a privacy guarantee is a relatively novel "bridge" between two fields.

---

> ### Author Rebuttal · Authors · 2026-03-28
>
> $\color{#2F80ED}{\text{Response to KQ1}}$
>
> We will clarify in the Introduction:
>
> In this paper, we use “inference-time privacy” to refer to privacy protection at the inference stage against leakage from released predictions (e.g., leakage of sensitive input attributes). That is, RP is a privacy notion for the inference stage.
>
> $\color{#2F80ED}{\text{Response to KQ2 and Limitations}}$
>
> To address the confusion, we will add the following first paragraph to the Threat Model section and the second paragraph to the Discussion section:
>
> Privacy is typically studied for training data, whereas input data at inference stage can also suffer privacy leakage. Examples include system prompt leakage in LLM inference and sensitive attribute leakage in personalized models. Moreover, traditional training-data privacy can also be exposed through the inference interface, such as confidence distributions or prediction changes exploited by model inversion attacks, so mitigating privacy leakage at inference stage can also protect training-data privacy.
>
> Prior work such as DP-CERT and PixelDP has established connections between privacy and robustness in practice. RP clarifies this connection by demonstrating, through two experiments, that prediction invariance around an input can be interpreted as inference-stage privacy. The mitigation effectiveness of RP is presented together with model utility in Figure 3 to demonstrate that privacy leakage can be mitigated more effectively while model accuracy remains higher when more noisy samples are used for a prediction. That is, if higher inference cost is allowed, both model privacy and model utility can be retained.
>
> $\color{#2F80ED}{\text{Response to KQ3}}$
>
> To clarify the confusion, the following will be added in Section 4, after the privacy metric required in [W2, reviewer mY9D]:
>
> An $\ell_2$ radius of $R$ is a geometric privacy quantity: under the released prediction, the queried input cannot be distinguished from other inputs within the $\ell_2$ ball of radius $R$ around it. For high-dimensional medical records, this means that dependence of the released prediction on a specific sensitive value can be blurred into a wider compatible range.
>
> For images, it means that the released prediction is compatible with many nearby candidate inputs in the input space. These nearby images can be viewed as noisy variants of the original image, making exact reconstruction through model inversion much more difficult.
>
> $\color{#2F80ED}{\text{Response to KQ4}}$
>
> The following will be added to the Discussion section to clarify the confusion (similar to that of KQ2, Reviewer mY9D):
>
> Simply adding noise to the output randomly perturbs the released prediction itself, and therefore obviously compromises model utility. In contrast, RP instantiated with randomized smoothing uses multiple noisy versions of the input for one prediction and aggregates those predictions, so the final released prediction is stabilized rather than randomly perturbed.
>
> A DP-trained model primarily protects training data by adding noise during training, but it can still suffer from model inversion, since representative training-data information may still be memorized and extracted. RP is a post-hoc way to offer privacy at inference stage without affecting the original model, and to mitigate privacy attacks that leverage leakage through the inference interface (e.g., model inversion).

---

> > ### Author Rebuttal · Reviewer_wJUt · 2026-04-03
> >
> > I appreciate the detailed rebuttal and have considered your responses with care. The direction of the work remains interesting, and the proposed clarifications are welcome. That said, after reflecting on the full set of revisions, my overall evaluation has not changed. I feel the paper still requires further development before it is ready for acceptance.

---

### Official Review · Reviewer_JqyH · 2026-03-11

**Soundness:** 2
**Presentation:** 2
**Significance:** 2
**Originality:** 2
**Overall Recommendation:** 4
**Confidence:** 3

**Summary:**

This paper proposes robust privacy, a new inference-time privacy concept inspired by certified robustness. The main idea is to ensure that model outputs remain provably invariant within an r-radius neighborhood around an input sample. Under this guarantee, an adversary cannot accurately reconstruct the input from the model predictions. In particular, the paper discusses defenses against both attribute inference attacks and model inversion attacks. The authors implement these attacks and evaluate the proposed mechanism, and the results validate its effectiveness.

**Compliance With Llm Reviewing Policy:**

Affirmed.

**Final Justification:**

The rebuttal has addressed my concerns.

**Key Questions For Authors:**

1. Could you please clarify the threat model and provide some realistic attack scenarios to justify it?
2. Could you further elaborate on and justify the technical novelty of this work? At present, many of the technical solutions appear to be adapted from the certified robustness literature.

**Limitations:**

Yes

**Strengths And Weaknesses:**

Strengths:
1. Interesting problem: The problem motivation is clear, and the fundamental idea of providing an inference-time privacy guarantee framework is both interesting and important.
2. Clear paper writing: The paper’s presentation logic is clear and easy to follow.
3. Extensive experiments: The paper conducts extensive experiments on both attacks and discusses the results in two separate sections, which clearly demonstrate the effectiveness of the proposed framework.

Weaknesses:
1. Threat model needs further clarification: The threat model is not clearly articulated, and I am confused about the adversary’s capabilities and knowledge. In particular, it is unclear how an adversary can both query the model with input samples and observe the corresponding outputs. Under this setting, the adversary appears to already possess the samples, which makes the need for launching a privacy attack unclear. The authors may need to provide a practical attack scenario to better justify this threat model.
2. The attack settings are unclear: The attribute inference attack and model inversion attack appear to be technically similar, except that the attribute inference attack assumes the adversary has additional knowledge of other attributes. The paper may need to further clarify how these two attacks differ, especially since the experimental section evaluates them separately.
3. Not enough technical novelty: The paper introduces the concept of robust privacy. However, the mechanism to achieve robust privacy is still the existing method used to achieve certified robustness. Therefore, the paper seems to be a direct adaption of certified robustness without too much technical new content. This makes the technical contribution unclear.

---

> ### Author Rebuttal · Authors · 2026-03-28
>
> $\color{#2F80ED}{\text{Response to W1, W2, and KQ1}}$
>
> We will revise the Threat Model section to make this clearer:
>
> For Sensitive Attribute Inference, the adversary observes the released prediction for a target user and combines it with side information, namely $x_{-1}$ (all non-sensitive attributes other than the target sensitive attribute $x_1$), to infer the unknown sensitive attribute $x_1$. This is the standard assumption in attribute inference attacks. For example, an adversary may observe a user’s recommendation result (e.g., whether insulin is recommended) in a public setting, such as on a subway or in an elevator, and combine it with publicly available non-sensitive attributes to infer the unknown sensitive attribute.
>
> For Model Inversion, the adversary is a label-only black-box querier. The attacker has no access to private training data; instead, it submits synthetic inputs, observes prediction changes, and iteratively updates the synthetic input to approximate representative training data for a target class.
>
> Sensitive Attribute Inference aims to infer a specific unknown sensitive attribute of an inference-stage input from the released prediction. Model Inversion aims to reconstruct a representative sample of a target class that is memorized from the training distribution at training time, through prediction changes of synthetic inputs observed from the inference interface.
>
> $\color{#2F80ED}{\text{Response to W3 and KQ2}}$
>
> We will address this concern by clarifying in the Introduction and contribution:
>
> The primary novelty is a new formalization of inference-stage privacy. Privacy is typically studied for training data, whereas input data at inference stage can also suffer privacy leakage. Examples include system prompt leakage in LLM inference and sensitive attribute leakage in personalized models. Moreover, traditional training-data privacy can also be exposed through the inference interface, such as confidence distributions or prediction changes exploited by model inversion attacks.
>
> Robust Privacy provides inference-stage privacy against both queried-input leakage and training-data leakage through the inference interface. Specifically, RP leverages the Attribute Privacy Enhancement effect to blur the dependence of the released prediction on a specific sensitive value into a wider compatible range, and masks the fine-grained input-output dependence signal at inference stage to mitigate model inversion.
>
> Therefore, the contribution of this paper is a new inference-stage privacy formalization and its evaluation in mitigating leakage of input sensitive attributes and training-data information through the inference interface.

---

> > ### Author Rebuttal · Reviewer_JqyH · 2026-04-02
> >
> > Thank you for your response. The rebuttal has addressed my concerns regarding the threat model. I have one quick follow-up question on inference-time privacy. Differential privacy was originally developed for privacy-preserving data analysis rather than specifically for model training, and it may also be applicable at the inference stage. In this context, it could protect released outputs or user queries through randomized mechanisms. Could you briefly discuss whether inference-time DP is suitable for your setting, and clarify how it differs from your method?

---

> > > ### Author Response · Authors · 2026-04-03
> > >
> > > Thanks for the follow-up question. The question can be answered by a combination of responses to W3 and KQ2 of Reviewer mY9D:
> > >
> > > Differential Privacy (DP) provides privacy guarantees by masking the contribution of any individual record in a dataset with calibrated noise, so DP is typically applied to protect the privacy of the training dataset. Robust Privacy, when instantiated with randomized smoothing, does not calibrate noise for individual records in a dataset, but instead adds noise to multiple copies of an input and aggregates the final noisy predictions to certify geometric indistinguishability around the input under the released prediction, thus claiming inference-stage privacy.
> > >
> > > Therefore, DP protects individual records in a dataset by masking their contribution to the final result, while RP protects an individual inference input by masking the fine-grained input-output relationship.
> > >
> > > Applying DP at inference stage typically means adding noise to the output. However, simply adding noise to the output randomly perturbs the released prediction itself, and therefore obviously compromises model utility. In contrast, RP instantiated with randomized smoothing uses multiple noisy versions of the input for one prediction and aggregates those predictions, so the final released prediction is stabilized rather than randomly perturbed.
> > >
> > > Therefore, inference-stage DP can in principle be used in our setting, but for the reasons above, it is not the recommended mechanism here.

---

### Official Review · Reviewer_mY9D · 2026-03-12

**Soundness:** 3
**Presentation:** 3
**Significance:** 3
**Originality:** 3
**Overall Recommendation:** 4
**Confidence:** 3

**Summary:**

The paper introduces Robust Privacy (RP), an inference-time privacy notion inspired by certified robustness. RP provides an indistinguishability guarantee within an R-bounded neighbourhood around each input. This mitigates inference-time side-channel leakage about the input through released predictions. A larger R corresponds to stronger privacy at inference time. The paper then translates input-level invariance under Robust Privacy into an attribute-level privacy effect, formalized as Attribute Privacy Enhancement (APE).

**Compliance With Llm Reviewing Policy:**

Affirmed.

**Final Justification:**

Thank you for the detailed rebuttal, however, my overall evaluation remains unchanged. The paper still requires further improvements before it is ready for acceptance.

**Key Questions For Authors:**

- Can the authors justify why existing approaches combining DP training with robustness techniques would not already provide the protections claimed by RP, while also offering formal privacy guarantees?

- Have you compared against simple output perturbation baselines (e.g., adding Gaussian noise to logits, random label flipping) for the model inversion experiment?

**Limitations:**

The above weaknesses.

**Strengths And Weaknesses:**

Strengths
- The paper identifies a real and important problem: privacy leakage, i.e., the exposure of sensitive user information, that can occur during the prediction phase (inference time) in personalized ML systems.
- The results are interesting and demonstrate a meaningful privacy–utility trade-off.


Weaknesses
- The paper fails to adequately engage with the existing literature on inference-time privacy, particularly DP-CERT (Wu et al., TMLR 2024), Shredder (Mireshghallah et al., ASPLOS 2020), PixelDP (Lecuyer et al., 2019), and PATE (Papernot et al., ICLR 2018), all of which address closely related problems with formal guarantees.
- The paper does not provide formal privacy metrics, such as bounds on the mutual information between the input and output, that would allow practitioners to reason about the actual privacy afforded.
- Section 7.3 presents differential privacy (DP) and robust privacy (RP) as complementary yet independent, noting that DP protects training data while RP protects inference-time inputs. However, this characterization is somewhat misleading, as prior work, such as DP-CERT, has demonstrated that DP and certified robustness are interdependent in practice. The paper would benefit from comparing its approach with existing methods that explicitly link DP to certified robustness and from clarifying the new insights or advantages it offers beyond previous works.

Minor Note: The paper uses the term “model inversion attacks (MIAs)” with the acronym MIA, which typically refers to “membership inference attacks” in the privacy literature. This creates a confusion and should be clarified.

---

> ### Author Rebuttal · Authors · 2026-03-29
>
> $\color{#2F80ED}{\text{Response to W1}}$
>
> We will add a discussion of these works in the Related Work section to address this gap:
>
> PixelDP connects differential privacy to certified robustness and uses DP-inspired noise injection (e.g., noise added to the input) to derive robustness certificates.
>
> DP-CERT jointly obtains privacy and robustness by integrating augmentation multiplicity into DP training: gradients from augmentations of each sample are averaged before clipping and noising to preserve the DP guarantee of DPSGD, while augmentation-based training also improves robustness.
>
> PATE is a training-time privacy framework in which a student model is trained on noisy aggregated outputs from teacher models; hence, the sensitive training data of the teacher models is protected.
>
> Shredder protects inference privacy in split edge-cloud inference by learning additive noise distributions for transmitted intermediate results, reducing the information content of the communicated representations while preserving inference accuracy.
>
> RP formalizes robustness as an inference-stage privacy guarantee under the released prediction. RP mitigates both sensitive-attribute leakage of the input from released predictions and training-data leakage through the inference interface.
>
> $\color{#2F80ED}{\text{Response to W2}}$
>
> We agree that the current draft does not make the formal privacy metric of RP sufficiently explicit. In our framework, the robust radius $R$ is precisely the formal privacy metric provided by Robust Privacy. We will add a formal definition of the Robust Privacy Metric in Section 4 and then reference this definition in the two experimental sections:
>
> Given a model $f:\mathcal{X}\to\mathcal{Y}$, an input $x\in\mathcal{X}$, and an $\ell_p$ norm, define $R$ as a robust radius such that, for all inputs $x'$ with $\|x'-x\|_p\le R$, the prediction remains unchanged. That is, $R$ is a robust radius around $x$ within which observing the released prediction cannot distinguish $x$ from neighboring inputs. A larger $R$ indicates stronger Robust Privacy.
>
> $\color{#2F80ED}{\text{Response to W3}}$
>
> Besides the response to W1, which expands related work such as DP-CERT, we will revise Section 7.3 to include the following content:
>
> Differential Privacy (DP) provides privacy guarantees by masking the contribution of any individual record in a dataset with calibrated noise, so DP is typically applied to protect the privacy of the training dataset. Robust Privacy, when instantiated with randomized smoothing, does not calibrate noise for individual records in a dataset, but instead adds noise to multiple copies of an input and aggregates the final noisy predictions to certify geometric indistinguishability around the input under the released prediction, thus claiming inference-stage privacy.
>
> Therefore, DP protects individual records in a dataset by masking their contribution to the final result, while RP protects an individual inference input by masking the fine-grained input-output relationship.
>
> However, DP and RP are not independent: prior work such as PixelDP has connected differential privacy and robustness, which is reinterpreted in this study as Robust Privacy. Beyond this reinterpretation, our contribution is to demonstrate through two experiments that RP mitigates leakage of input sensitive attributes and training-data information through the inference interface.
>
> $\color{#2F80ED}{\text{Response to MN1}}$
>
> We will clarify in the Introduction and Section 7.4: “MIA” in this paper refers specifically to Model Inversion Attacks.
>
> $\color{#2F80ED}{\text{Response to KQ1}}$
>
> We will clarify in the Introduction and contribution:
>
> Although approaches combining DP training with robustness techniques may already provide inference-time protection for queried inputs, such protections are typically introduced and analyzed through the lens of training-data privacy and robustness, rather than being explicitly formalized as an inference-time privacy guarantee for the input under the released prediction. Robust Privacy formalizes inference-time privacy and demonstrates its effect in two scenarios: sensitive-attribute inference and model inversion.
>
> $\color{#2F80ED}{\text{Response to KQ2}}$
>
> To clarify the reviewer’s confusion (similar to that of W4, Reviewer wJUt), we will add the following to the Discussion section:
>
> Simply adding noise to the output randomly perturbs the released prediction itself, and therefore obviously compromises model utility. In contrast, RP instantiated with randomized smoothing uses multiple noisy versions of the input for one prediction and aggregates those predictions, so the final released prediction is stabilized rather than randomly perturbed. Moreover, we use the unprotected model as the baseline because it provides the reference point for isolating the mitigation effect introduced by RP relative to the original classifier.

---

### Decision · Program_Chairs · 2026-04-30

**Decision:**

Reject

**Comment:**

Reviewers agreed that this paper explores an interesting direction and that connecting certified robustness to inference-stage leakage is potentially valuable. The paper offers some encouraging empirical observations on sensitive-attribute inference and model inversion.

However, following the reviewer discussion, I do not think the current version makes a sufficiently compelling case for acceptance. The main concern is that the paper’s central claim -- interpreting local prediction invariance within an input neighborhood as privacy -- was not established strongly enough as a meaningful privacy notion in contrast to simply a way to achieve smoothing or reduced sensitivity. Several reviewers remained unconvinced that the proposed notion captures privacy in a principled way rather than a utility–sensitivity tradeoff. In addition, the paper does not engage in depth with closely related prior work connecting privacy and robustness, and the empirical comparison to differential privacy and simpler noise-based baselines is not strong enough to support the claimed advantages. In particular, if we accept the new notion as a privacy notion, the privacy-utility trade-off in this case (tunable through the radius of input uncertainty R) could be as bad as the one if we were to use input/output DP perturbation (tunable through \varepsilon) and it is not clear what parameters provide comparable protection. The experimental scope also remains limited relative to the paper’s framing.

For these reasons, the paper cannot be accepted at this time. I do think the direction is promising, and a revised version with a sharper formalization, clearer positioning relative to prior privacy notions, and stronger experimental evaluation could be interesting for a future venue.